# RhoA/ROCK Pathway Is Upregulated in Experimental Autoimmune Myocarditis and Is Inhibited by Simvastatin at the Stage of Myosin Light Chain Phosphorylation

**DOI:** 10.3390/biomedicines12030596

**Published:** 2024-03-07

**Authors:** Monika Skrzypiec-Spring, Maciej Kaczorowski, Alina Rak-Pasikowska, Agnieszka Sapa-Wojciechowska, Krzysztof Kujawa, Agnieszka Żuryń, Iwona Bil-Lula, Agnieszka Hałoń, Adam Szeląg

**Affiliations:** 1Department of Pharmacology, Wroclaw Medical University, 50-345 Wroclaw, Poland; adam.szelag@umw.edu.pl; 2Department of Clinical and Experimental Pathology, Wroclaw Medical University, 50-556 Wroclaw, Poland; maciej.kaczorowski@umw.edu.pl (M.K.); agnieszka.halon@umw.edu.pl (A.H.); 3Division of Clinical Chemistry and Laboratory Hematology, Department of Medical Laboratory Diagnostics, Wroclaw Medical University, 50-556 Wroclaw, Poland; alina.rak-pasikowska@umw.edu.pl (A.R.-P.); agnieszka.sapa-wojciechowska@umw.edu.pl (A.S.-W.); iwona.bil-lula@umw.edu.pl (I.B.-L.); 4Statistical Analysis Centre, Wroclaw Medical University, 50-368 Wroclaw, Poland; krzysztof.kujawa@umw.edu.pl; 5Department of Histology and Embryology, Collegium Medicum, Nicolaus Copernicus University, 85-067 Bydgoszcz, Poland; azuryn@cm.umk.pl

**Keywords:** immune system, myocarditis, molecular pathology, RhoA/ROCK pathway, simvastatin, metalloproteinases

## Abstract

Many studies have proven the involvement of the RhoA/ROCK pathway in autoimmune and cardiovascular diseases and the beneficial effects of its downregulation. Here, we examined whether the effect of simvastatin on experimental autoimmune myocarditis (EAM) may be through targeting the Ras homolog family member A/Rho-associated coiled-coil containing kinases (RhoA/ROCK) pathway and whether previously shown downregulation of metalloproteinase 9 (MMP-9) could be associated with MLC phosphorylation. Two doses of simvastatin were administered to experimental rats with autoimmune myocarditis by gastric gavage for 3 weeks, at the stage of development of the inflammatory process. Immunohistochemical staining for RhoA and ROCK1 was evaluated semi-quantitatively with H-score. The RhoA staining showed no significant differences in expression between the groups, but the ROCK1 expression was significantly upregulated in the hearts of the EAM group and was not downregulated by simvastatin. The Western blotting analysis of the last downstream product of the RhoA/ROCK axis, phosphorylated myosin light chain (phospho-MYL9), revealed that protein content increased in EAM hearts and it was prevented by the highest dose of simvastatin. Our findings suggest that the RhoA/ROCK pathway is upregulated in EAM, and simvastatin in EAM settings inhibits the RhoA/ROCK pathway at the stage of phosphorylation of myosin light chains and provides a new insight into the molecular pathology of autoimmune myocarditis.

## 1. Introduction

Autoimmunity has been recognized as one of the main causes of myocarditis [1]. Epidemiological data provide evidence of an increased incidence of autoimmune diseases in Western societies in recent decades [2]. Autoimmune diseases such as systemic lupus erythematosus, Sjögren’s syndrome, vasculitis, ulcerative colitis, and polymyositis may be associated with giant cell myocarditis [3,4]. This type of myocarditis has a low incidence but poor prognosis despite maximum therapy. Another subtype of myocarditis associated with systemic autoimmune disease is eosinophilic myocarditis [5,6]. It occurs in 50–60 percent of peripheral eosinophilia, mainly in men. Cardiac involvement also occurs in 2 to 5 percent of systemic sarcoidosis [7,8,9,10]. In clinical trials, myocarditis was reported in approximately 9 percent of patients with systemic lupus erythematosus, but postmortem analyses showed a high incidence of subclinical disease, as myocarditis was found in 57 percent of cases [2]. Autoimmunity has also been shown to maintain inflammation and disease progression in viral myocarditis [2]. Because many patients with myocarditis may develop dilated cardiomyopathy despite standard optimal treatment, the pathomechanisms associated with autoimmune disease are being investigated and new ways to prevent systolic dysfunction are being sought. In recent years, the important role of Ras homolog family member A/Rho-associated coiled-coil containing kinases (RhoA/ROCK) in the regulation of immune responses has been discovered. Rho-kinases belong to serine/threonine protein kinases. They have been identified as downstream RhoA effectors that mediate calcium sensitization. The best characterized RhoA effector is ROCK, which exists in two isoforms: ROCK2 and ROCK1. The main substrate of Rho-kinases is myosin light chain (MLC) phosphatase, which is responsible for the dephosphorylation of the light chains of myosin II. Many studies have suggested the importance of the RhoA/ROCK pathway in autoimmunity. Manresa-Arraut et al. demonstrated that genetic deletion of *RhoA* in T cells reduces the severity of experimental autoimmune encephalomyelitis [11]. Sun et al. observed a similar neuroprotective effect with the use of the selective Rho-kinase inhibitor fasudil [12]. Furthermore, fasudil was found to attenuate the development of spontaneous arthritis and systemic lupus erythematosus in transgenic mouse models [13]. In the heart, the RhoA/ROCK pathway has been mainly studied in the context of cardiac fibrosis and apoptosis. Excessive ROCK activation is the cause of many cardiovascular pathologies, such as cardiac hypertrophy, apoptosis, and systemic and pulmonary hypertension [14]. It was also shown that *ROCK2* mRNA, but not *ROCK1* mRNA, was significantly increased in viral myocarditis. Moreover, the RhoA/ROCK pathway inhibitor fasudil increased survival, attenuated myocardial necrosis, and reduced viral replication and ROCK2 and IL-17 expression in infected hearts [15]. The involvement of the RhoA/ROCK pathway in the development of experimental autoimmune myocarditis (EAM) was only investigated by Tkacz et al. [16]. The authors showed that ROCK1 haploinsufficiency does not protect against developed CD4+ T-mediated heart inflammation, suggesting that the RhoA/ROCK pathway is not critically involved in the development of EAM. To our knowledge, there are no other scientific studies on the RhoA/ROCK pathway in EAM pathology and the effect of RhoA/ROCK pathway kinase inhibitors on its intensity. We have previously shown that the simvastatin treatment reduces mechanical cardiac dysfunction, myofilament damage, and troponin I degradation in EAM by inhibiting metalloproteinase 9 (MMP-9) activity [17]. We also showed that simvastatin has a protective effect on cardiac systolic function in an acute ischemia–reperfusion model by inhibiting the RhoA pathway and is independent of reductions in metalloprotease 2 (MMP-2) activity [18]. On the other hand, simvastatin, lovastatin, and atorvastatin have been shown to inhibit MMP-2 and MMP-9 activation by downregulating myosin phosphatase target subunit 1 (MYPT1) and MLC, which are downstream substrates of the RhoA/ROCK pathway [19]. Since simvastatin is an inhibitor of the RhoA/ROCK pathway, the present study was designed to investigate whether the cardioprotective properties of simvastatin in EAM could be mediated, at least in part, by targeting the RhoA/ROCK pathway and whether the previously demonstrated downregulation of MMP-9 could be associated with MLC phosphorylation.

## 2. Materials and Methods

### 2.1. Animals

This study protocol was approved by the First Local Bioethics Committee For Animal Experiments at the Wrocław Institute of Immunology and Experimental Therapy of the Polish Academy of Sciences (approval No. 25/2012). The study was carried out in compliance with the ARRIVE guidelines. All methods were carried out in accordance with relevant guidelines and regulations. Twenty-three female Lewis rats, 6–8 weeks old, purchased from AnimaLab (Poznan, Poland), were used for the experiment. The animals were randomly assigned to one of 4 groups and housed together under the same conditions.

### 2.2. Induction of Active EAM and Protocol of Experiment

The active EAM was induced using the protocol described by Kodama et al. Briefly, rats were injected subcutaneously into one hind footpad with an antigen-adjuvant emulsion prepared by emulsifying purified porcine cardiac myosin (Sigma-Aldrich, Poznań, Poland) with an equal volume of complete Freund’s adjuvant (Difco, Warsaw, Poland) supplemented with Mycobacterium tuberculosis strain H37Ra (Difco) to a final concentration of 5 mg/mL. On days 0 and 7, 15 rats were injected with 0.1 mL of the emulsion into the footpad. The 10 EAM-induced rats were divided into two groups (*n* = 5 each): treated with simvastatin 30 mg/kg daily (EAM 30) and simvastatin 3 mg/kg daily (EAM 3) in 0.5 percent methyl-cellulose. Simvastatin was administered orally by gastric gavage for 3 weeks, from day 0 to day 21 (Sigma-Aldrich). Other EAM (EAM, *n* = 5)-induced rats and unimmunized rats (C, *n* = 8) received the vehicle orally by gastric gavage for 3 weeks from day 0 to day 21. Since the current study is part of a project and the results of its first part have already been published [17], due to the insufficient amount of tissue remaining after the analyses in the previous part, group C in the Western blot analysis consisted of 4 or 5 samples.

### 2.3. Sample Collection

All stages of the experiments were carried out in such a way as to avoid the animals suffering. On day 21, all animals were anesthetized with ketamine (10 mg/kg body weight) and then sacrificed by decapitation. Hearts were collected and divided into two parts. One part of each heart was quickly deep-frozen and pulverized at liquid nitrogen temperature and stored at −80 °C until further analysis. The remainder of each heart was fixed in 4 percent buffered formalin (Chempur, Piekary Śląskie, Poland). The samples were then rinsed under running water, dehydrated in increasing concentrations of ethyl alcohol (Stanlab, Lublin, Poland), passed through Neo-Clear intermediate liquid (Merck, Darmstadt, Germany), and embedded in paraffin blocks in such a way to enable the visualization of axial sections of analyzed hearts in subsequently prepared microscopic slides.

### 2.4. Immunohistochemical Studies

Paraffin blocks with embedded tissues of rat hearts were cut with a microtome into 4 μm thick sections that were mounted on silanized glass slides (Agilent DAKO, Santa Clara, CA, USA). Dewaxing, rehydration and heat-induced epitope retrieval were processed in a PT Link Pre-Treatment Module for Tissue Specimens (DAKO) through a 20 min incubation at 95 °C with the use of High pH EnVision Target Retrieval Solution (Agilent DAKO). Automated staining with anti-RhoA (mouse monoclonal, clone 26C4; dilution 1:50; Santa Cruz Biotechnology; Dallas, TX, USA), anti-ROCK1 (rabbit monoclonal, clone EPR638Y, dilution 1:100; Abcam, Cambridge, UK), was carried out in Autostainer Link 48 (DAKO) with DAB used as a chromogen. Human brain and testis tissues were used as positive controls for the detection of RhoA and ROCK1, respectively. The evaluation of immunohistochemical stains was performed by a surgical pathologist. RhoA and ROCK1 stains were analyzed in a semi-quantitative manner with the use of an H-score. H-score is a standardized method to evaluate protein expression in tissue sections stained with immunohistochemistry. This scale combines two parameters of an immunohistochemical reaction: the intensity of staining and the percentage of reactive tissue. In our experiment, the intensity of staining in heart muscle cells ranging from 0 to 3 (0: no staining; 1: weak staining; 2: moderate staining; 3: strong staining) was multiplied with the percentage of the tissue section exhibiting a respective reaction intensity. Then, products obtained for individual intensities observed in a single specimen were summed to obtain a final H-score ranging from 0 to 300.

### 2.5. Western Blot

One part of each heart was frozen in liquid nitrogen for 5 min and then crushed and grinded using a porcelain pestle and mortar. A portion of tissue powder was mixed with homogenization buffer (50 mmol/L Tris-HCl, 150 mmol/L NaCl, 0.1 percent Triton X-100, pH 7.4) containing Protease Inhibitors Cocktail Set III (Sigma-Aldrich, St. Louis, MO, USA) in 1:5 proportion (*w*:*v*), 1 percent of Phosphatase Inhibitor Cocktail 2, and 1 percent of Phosphatase Inhibitor Cocktail 3 (Sigma-Aldrich, St. Louis, MO, USA). Next, samples were homogenized on ice by a Pellet Pestle^®^ Motor (Kimble Kontes, Vineland, NJ, USA). Samples were centrifuged (5 min, 14,000 rpm, 4 °C) and the protein concentration in supernatants was measured using Bradford Protein Assay (Bio-Rad, Hercules, CA, USA). To detect the phospho-MYL9 in tissue homogenates, Western blotting was used. Briefly, the first step of blotting was SDS-PAGE electrophoresis (120 V, 20 °C). The 10% gels were prepared on the basis of TGX Stain-Free FastCast Acrylamide Kit 10 percent (BioRad, Hercules, CA, USA). The optimal amount of sample to be loaded on the gel was determined by assessment of signal intensity dependence on protein concentration. All samples containing 40 µg of protein were mixed with 4× Laemmli Sample Buffer (BioRad, Hercules, CA, USA) with the addition of beta-mercaptoethanol (1:10; *v*:*v*). Samples for ROCK1, MLC1, and alpha-tubulin detection were heated at 100 °C for 5 min. One of the samples was loaded in each gel to ensure reproducibility and served as a standard for calculations. After separation in the gels, the proteins were electrotransferred (by wet transfer) to the nitrocellulose membranes (50 V, 45 min). The membranes were incubated overnight at 4 °C with primary antibodies as follows: (1) the rabbit polyclonal IgG antibody against myosin light chain 2 (1:1000) (PA5-17727, Invitrogen, Rockford, IL, USA); (2) rabbit polyclonal IgG against ROCK1 (1:1000) (PA5-22262, Invitrogen, Rockford, IL, USA); (3) rabbit polyclonal IgG against MLC1 (1:1000) (ab94730, Abcam, Cambridge, UK); and (4) mouse monoclonal IgG against alpha-tubulin (1:5000) (ab7291, Abcam, Cambridge, UK). The secondary goat-anti-rabbit conjugated with HRP and goat-anti-mouse conjugated with HRP (both from BioRad, Hercules, CA, USA) were diluted in 1:3000. To visualize protein bands on the membrane, the chemiluminescence assay (ClarityTM Western ECL Substrate, BioRad, Hercules, CA, USA) was used. Membranes were scanned by the ChemiDocTM XRS+ System with Image LabTM Software v.5.2 for data analysis. A relative quantity of the proteins was expressed in arbitrary units (AU) and calculated using one of the samples serving as a standard with a given value of 1 AU in each gel. Protein loading was normalized to alpha-tubulin. The Precision Plus Protein™ All Blue Standards (BioRad, Hercules, CA, USA) served as a protein molecular weight standard. 

### 2.6. Statistical Analysis

All data presented in the figures are expressed as mean ± 95%CI. Statistical analysis was performed using one-way parametric ANOVA with Welch correction in some cases when the variance homogeneity assumption was not met. Variance homogeneity was checked using the Levene test. As the sample size per group was small, checking normal data distribution in groups was significantly limited. The adjusted coefficients of determination R^2^ (adj-R^2^) were used as goodness-of-fit measures of models. Adj-R^2^ represents the proportion of variability in the dependent variable that may be explained by the given set of independent variables. Adjusting R^2^ adjusts this value based on the number of predictors in the model. Two post hoc tests were used: HSD Tukey’s test when the between-group variance was equal, and the Dunnett’s T3 test otherwise. *p* values < 0.05 were considered statistically significant. Statistica 13 (TIBCO Software Inc., Palo Alto, CA, USA, 2017) and IBM SPSS 28 (IBM Corp., Armonk, NY, USA, Released 2021) were used for statistical analysis.

## 3. Results

### 3.1. RhoA Expression in Heart Tissue Was Not Upregulated and Not Affected by Simvastatin

To test whether the previously demonstrated cardioprotective properties of simvastatin in EAM could be mediated, at least in part, by targeting the RhoA/ROCK pathway, we performed immunohistochemistry and analyzed RhoA staining in a semiquantitative manner using the H-score. The expression of RhoA, summarized in Figure 1 and Figure 2a–d, was observed in the cytoplasm of cardiomyocytes of the rat hearts. The highest expression levels were observed in the EAM group, while the lowest were in the EAM 30 group; however, the differences were not statistically significant.

### 3.2. Expression of ROCK1 Was Upregulated EAM but Not Affected by Simvastatin

ROCK1 staining was then assessed in a semi-quantitative manner using the H-score scale. At least weak and focal cytoplasmic expression of ROCK1 in cardiomyocytes was detected in all samples, although it was minimal in samples from the C group. ROCK1 expression was significantly upregulated in the hearts of all EAM groups compared with control cases, *p* < 0.001 in ANOVA post hoc Tukey’s test (Figure 3 and Figure 4a–d).

### 3.3. The Western Blotting Analysis for ROCK1 Was Not Reliable

Due to the non-stoichiometric nature of immunohistochemistry, in the next step of the experiment, the ROCK1 protein content was assessed using the Western blot method. Western blot analysis did not show statistical significance in terms of the increase in ROCK 1 content in EAM; however, the coefficient of determination R^2^ indicated that only 17.57% of the variability was the result of group membership (Figure 5a,b).

### 3.4. Phosphorylated Myosin Light Chain Protein Content Was Upregulated in EAM and the Treatment with Simvastatin Prevented the EAM-Induced Increase in Phosphorylation of MLC

In the next step of the experiment, the content of the phosphorylated myosin light chain (phospho-MYL9) protein was assessed by Western blotting to check whether the downregulation of MMP-9 in simvastatin-treated EAM hearts, which we previously demonstrated, could be associated with MLC phosphorylation. Phospho-MYL9 protein content was upregulated in the EAM group, with the difference being statistically significant between groups C and EAM and C and EAM 3 (*p* < 0.05, respectively) indicating that simvastatin at a dose of 3 mg/kg did not inhibit the inflammation-induced MLC phosphorylation. However, in the EAM 30 group, phospho-MYL9 protein content was comparable to that of the C group, indicating that simvastatin at a dose of 30 mg/kg prevented a significant increase in MLC phosphorylation (Figure 6a,b).

### 3.5. Myosin Light Chain Protein Content Was Not Upregulated in EAM

No significant differences in MLC1 content were observed between groups (Figure 7a,b).

## 4. Discussion

Autoimmunity is one of the main causes of myocarditis which may progress to heart failure. Under physiological conditions, the heart muscle is not deprived of its own immune components but contains many immune cells [20,21,22,23]. The recruitment of inflammatory cells into the myocardial interstitium is promoted in various myocardial diseases, including ischemia–reperfusion injury, myocardial infarction, hypertension, and myocarditis. In autoimmune myocarditis, after the initiation of autoreactive sensitization, both innate and adaptive immune cells migrate into the interstitial space of the heart and contribute to the initiation and maintenance of T cell autoreactivity. In T and B cells, which are the effector cells of the adaptive immune system, Rho signaling is crucial for activation and migration. It plays an important role in responding to chemokines, cytokines, and growth factors released from both innate and adaptive immune cells [24]. On the other hand, it serves as one of the key proteins involved in stress-mediated cardiomyocyte signal transduction [23].

Statins are used as inhibitors of cholesterol biosynthesis. They act by reversibly inhibiting HMG-CoA reductase, which is the rate-limiting enzyme of cholesterol biosynthesis and catalyzes the conversion of HMG-CoA to mevalonic acid. However, the overall beneficial effects of statins are greater than would be expected from lipid lowering alone, suggesting a cholesterol-independent, “pleiotropic” effect of statins. By mainly inhibiting the synthesis of L-mevalonic acid, statins also prevent the synthesis of other isoprenoid intermediates of the cholesterol biosynthetic pathway. These intermediates, such as farnesyl pyrophosphate (FPP) and geranylgeranyl pyrophosphate (GGPP), are involved in the post-translational modification of many proteins, including the small guanosine triphosphate (GTP)-binding protein Ras and Ras-like proteins such as Rho or Rac [25,26]. Inhibition of Rho and its downstream target, Rho kinase, is largely responsible for some of the pleiotropic effects of statins. However, statins may also exert their beneficial effects by influencing other metabolic pathways. There are several mechanisms that may have a beneficial effect on the course of autoimmune myocarditis. These include the Rac1-mediated inhibition of NADH oxidase activity and downregulation of angiotensin AT1 receptor expression and inhibition of cardiac hypertrophy through an antioxidant mechanism involving the inhibition of Rac1 geranylgeranylation [26,27]. Recent data indicate that in the case of myocardial failure, the expression of the membrane proteins Rac1 and p47phox is increased, and there is also an increase in Rac1-GTPase activity, which may resemble the basic mechanisms of increased oxidase activity. The inhibition of angiotensin II-induced oxidative stress and cardiac hypertrophy by states has been confirmed in rodents [26,28]. Additionally, inhibition of geranylgeranyltransferase or RhoA leads to an increase in eNOS expression [29,30,31]. Another possible mechanism for the beneficial effects of statins on autoimmune myocarditis involves the inhibition of major histocompatibility complex class II (MHC-II) expression on endothelial cells, monocytes, and macrophages through inhibition of the class II transactivator IV (CIITA) promoter, leading to a decrease in MHC-II-mediated T cell activation [32]. Additionally, statins reduce CD40 expression and CD40-related vascular cell activation [33]. The effects of statins on the autoimmune system are also related to binding to a novel allosteric site within β2 integrin function-associated antigen 1 (LFA-1), independent of mevalonate production [34]. LFA-1 is a member of the integrin family and plays an important role in leukocyte transport and T cell activation. Another impact is on the remodeling of the heart muscle by the inhibition of the activity of MMPs.

We have previously shown that MMPs and the RhoA/ROCK pathway inhibitor simvastatin have a protective effect on cardiac systolic function in an acute ischemia–reperfusion model by inhibiting the RhoA pathway independently of reducing MMP-2 activity [18]. We also showed that simvastatin exerted a cardioprotective effect in a model of EAM by inhibiting MMP-9 activation and the subsequent inhibition of troponin I degradation [17]. In particular, the severity of inflammation and heart enlargement were reduced in the group treated with simvastatin at daily doses of 3 and 30 mg/kg compared to the EAM group. No significant left ventricular enlargement was observed in the hearts of rats treated with simvastatin, while in the EAM group, ultrasonography showed a significant increase in left ventricular end-diastolic diameter. Less severe myofibril degradation was also observed in the hearts of simvastatin-treated rats compared to EAM. Inflammatory cell infiltrations present in the EAM group in the perimysium and between cardiomyocytes, also in the form of thick clusters, were less intense in the groups receiving simvastatin, especially in the group treated with a higher dose. The number of multinucleated giant cells was also the lowest in this group [17].

Although many studies have proven the involvement of the RhoA/ROCK pathway in the development of autoimmune and cardiovascular diseases, its involvement in the pathomechanism of autoimmune myocarditis has not been investigated except for one publication on this subject.

As RhoA activation and RhoA-dependent downstream signaling play important roles in stress-exposed cardiomyocytes and the immune response, upregulation of the RhoA/ROCK pathway may be expected in autoimmune myocarditis. Our results support this hypothesis, as in the present study we showed that EAM is characterized by increased ROCK1 expression and increased phospho-MYL9 content. Surprisingly, Tkacz et al. demonstrated that ROCK1 haploinsufficiency does not protect against EAM and concluded that the RhoA/ROCK pathway is not critically involved in EAM development [16]. This is inconsistent with our observations, but these are the only existing data regarding the RhoA/ROCK pathway in EAM that have not yet been confirmed by anyone. Moreover, in their study, the authors did not examine whether the RhoA/ROCK pathway is stimulated in the EAM and whether its inhibition under stimulation conditions affects the course of EAM, but only examined the effect of its complete inhibition (in mice with ROCK1 and ROCK2 haploinsufficiency) on the development fibrosis in EAM induced in the absence of ROCK1 and ROCK2. Therefore, this protocol does not fully reflect the conditions that occur in reality during EAM, and therefore the results obtained by Tkacz et al. may differ from ours. This is also because there are several studies showing that in other types of inflammation, the RhoA/ROCK pathway is stimulated and its inhibition has a clinically beneficial effect.

The involvement of the RhoA/ROCK pathway was confirmed in viral myocarditis by Ma et al. Using a weighted gene co-expression network analysis (WGCNA) based on gene expression profiles from mouse models at different stages of viral myocarditis, the authors predicted hub genes as the genes most commonly associated with the disease in both its acute and chronic stages. They identified five hub genes in the acute stage of viral myocarditis, among them the Rho GTPase activating protein 30 [35].

*ROCK2* mRNA has also been shown to be increased in viral myocarditis and suppressed by fasudil, an inhibitor of the RhoA/ROCK pathway. The use of a RhoA/ROCK pathway inhibitor led to prolonged survival with the attenuation of myocardial necrosis, viral replication, and expression of ROCK2 and IL-17 in infected hearts [15]. There are also studies that have shown that the RhoA/ROCK pathway is activated in other myocardial inflammatory conditions such as endotoxemia and the beneficial effects of inhibitors of this pathway on the course of this disease. Endotoxemia can lead to the expression of cardiac pro-inflammatory cytokines and subsequent myocardial inflammation and dysfunction [36]. Preau at al. showed that fasudil, a potent RhoA/ROCK inhibitor, prevented LPS-induced cardiac inflammation, oxidative stress, cytoskeleton disarray, and mitochondrial injury [37].

In our work, we not only showed that the RhoA/ROCK pathway is activated in the EAM, but we also showed that the use of an inhibitor of this pathway, simvastatin (at the dose of 30 mg/kg daily), causes its inhibition at the level of MLC phosphorylation. These results are consistent with observations obtained in viral inflammation by Preau et al. and Dai et al. and open the door to new therapeutic possibilities in autoimmune myocarditis.

Additionally, there are reports indicating that the inhibition of MMP-2 and 9 activation may occur through the inhibition of the RhoA/ROCK pathway, but this issue has not yet been widely studied in heart diseases, although metalloproteinases play an important role in their pathomechanism. We have previously shown that MMPs and the RhoA/ROCK pathway inhibitor simvastatin have a protective effect on cardiac systolic function in an acute ischemia–reperfusion model by inhibiting the RhoA pathway independently of reducing MMP-2 activity [18]. We also showed that simvastatin exerted a cardioprotective effect in a model of EAM by inhibiting MMP-9 activation and subsequently inhibiting troponin I degradation [17]. The current results together with the previous ones indicate that simvastatin in the autoimmune myocarditis model affects not only MMP-9 activity but also the activity of the RhoA/ROCK pathway. These results are consistent with those of Kim et al. who showed that statins significantly inhibited both MYPT1 and MLC phosphorylation and the activation of MMP-2 and MMP-9 (enhanced by TGF-beta 2) in human optic head astrocytes and may suggest the existence of a relationship between these two pathways [19].

Rho kinase mediates MLC activation through its phosphorylation and phosphatase inactivation. MLC activation is believed to be necessary for vesicular transport and the subsequent secretion of Cyclophilin A [38]. Cyclophilin A has been shown to increase the production and activation of MMP-9 [39,40,41]. Postulated mechanisms leading to MMPs activation by Cyclophilin A include increased free radical generation and increased expression of the extracellular matrix metalloproteinase inducer (EMMPRIN, basigin, or CD147) [40]. Our results indicate that the use of simvastatin in EAM leads to the inhibition of MLC phosphorylation. The results of our previously published studies indicate that simvastatin at the same dose in EAM reduces MMP9 activity. Our current and previous results, in the light of the studies cited above, may indicate that the link connecting these two pathways is Cyclophilin A. The inhibition of MLC phosphorylation by simvastatin may lead to a reduction in its secretion and, as a result, to a reduction in EMMPRIN expression and the production of free radicals, which may result in a reduction in MMP-9 activity. Thus, Cyclophilin A could serve as a molecular marker to support our findings and would offer a broader understanding of the pathway’s regulation. Further research is required to confirm this hypothesis, and our results provide a promising starting point for further research.

The dose of simvastatin 30 mg/kg/d, which had a beneficial effect on the course of EAM (demonstrated by us previously) and on the inhibition of the RhoA/ROCK pathway demonstrated in the current study, corresponds to the dose used in humans. The highest recommended dose of simvastatin in humans for the treatment of hypercholesterolemia is 80 mg daily due to the drug’s side effects. This dose corresponds to 1.1 mg/kg for a person weighing approximately 70 kg. Most studies in rodents used much higher doses on a body weight basis in order to induce the same physiological changes as in humans (due to the pharmacodynamic resistance of rodents to statins). The use of such high doses in rodents may be due to pharmacodynamic resistance to pharmacological effects because statins do not reduce cholesterol levels in rodents, even though they inhibit the mevalonate pathway in the liver [42]. Westwood et al. showed that the administration of simvastatin at a dose of 60 mg/kg/day for 43 days did not cause any liver necrosis. There were also no significant changes in the hearts of rats after the administration of a dose of 60–80 mg/kg/day of simvastatin [43]. In animal studies, a high oral dose of simvastatin corresponds to 50 to 100 mg/kg/day. In this study, consistent with previous studies in animals, including rats, simvastatin doses of 3 and 30 mg/kg/day were selected as low and intermediate daily doses [44]. Demonstrating the beneficial effect of simvastatin in EAM at a dose corresponding to the therapeutic dose in humans is another important aspect of our results.

However, our study has some limitations. First of all, ROCK1 stimulation was only demonstrated in immunohistochemical tests. Western blot analysis also showed an increase in ROCK1 content in EAM, but it was not statistically significant. However, the coefficient of determination indicated that less than 20% of the variability was the result of group membership, so this result cannot be considered as clearly contradicting the results of the analysis. Even more so, Western blotting analysis revealed that the last downstream product of the RhoA/ROCK axis, phosphorylated myosin light chain (phospho-MYL9) protein content, increased in EAM hearts and that it was prevented by the highest dose of simvastatin. But, obviously, further research is required to confirm our observations. Second, based on our data, we cannot directly determine the exact mechanism responsible for the inhibition effect of simvastatin on the RhoA/ROCK pathway. This requires further detailed research, as well as examining the relationship between the inhibition of the RhoA/ROCK pathway and the activation of MMPs by simvastatin.

Another limitation is that induction of the RhoA/ROCK system has been demonstrated in a rat model of autoimmune myocarditis and not in humans. This is why our findings need to be translated from animal models to human diseases. EAM is an established animal model that manifests as human myocarditis in various stages and resembles human myocarditis based on its histological characteristics [45]. Primary infiltrates in the acute phase are the same in patients and rodents and consist of approximately 80% of macrophages and T and B lymphocytes (10–15%) [45,46,47,48]. Autoantibodies against cardiac myosin occur both in human myocarditis and in experimental models [45,49,50]. Also, the disease course and progression from myocarditis to cardiomyopathy over time are similar between animal models and human disease but induce myocarditis in a rather non-physiologic way.

Rat heart physiology, contraction patterns, and energetics are similar to those of humans. Rodent and human myocardial tissue is also similar, although rodent cardiomyocytes contain predominantly alpha-myosin heavy chains (compared to beta-myosin heavy chains in humans) which are characterized by rapid ATPase activity, facilitating high heart rates and short cardiac cycles [51]. The small physical size of rats must also be taken into account because only a limited number of post-mortem analyses can be performed on the same rat due to the small size of the tissues. Also important is the fact that animal experimental models mainly involve small groups, which affects the power of such studies. All of this must be taken into account when translating the findings to clinical practice in humans. However, despite these limitations, EAM in rats has been widely used to gain insight into the pathophysiology of myocarditis at the molecular and cellular levels and to test the efficacy and safety of treatment.

The treatment of myocarditis, despite advances in diagnosis and pharmacology, is still a challenge. Despite the use of available treatment methods, it often progresses to heart failure. Therefore, the search for new therapeutic strategies that can protect against the development of heart failure is very important from the point of view of the prevention of heart failure.

Simvastatin is a drug which has been on the market for a long time as a treatment for lipid disorder and comprehensive information is available on its pharmacology, dose, and possible toxicity. Its pleiotropic properties, including its influence on the immune system, redox system, and heart muscle remodeling, revealed in other studies and discussed above, provide a theoretical basis for its use in the treatment of inflammation in order to prevent the development of heart failure. Our results indicating that its use in therapeutic doses in myocarditis by blocking the RhoA/ROCK pathway and the previously demonstrated blocking of MMP 9 activation and beneficial effect on the course of inflammation indicate that its repurposing may constitute a promising and very attractive option.

## 5. Conclusions

In summary, in light of our current findings, we can conclude that the RhoA/ROCK pathway is upregulated in EAM and treatment with simvastatin (30 mg/kg, daily) at the stage of development of the inflammatory process prevents the increase in the concentration of the last metabolite of this pathway, i.e., phospho-MYL9. Further research is needed to confirm these findings and reveal the mechanism responsible for this phenomenon, but our results, although preliminary, are of great importance as they represent one of the first attempts to assess the role of the RhoA/ROCK pathway in the pathogenesis of EAM. They provide a new insight into the molecular pathology of autoimmune myocarditis and also make an important contribution to the study of the relationship between two important pathogenetic pathways resulting from the activation of MMP-9 and RhoA. This opens up new perspectives in understanding the pathophysiology of EAM and exploring new therapeutic options.

## Figures and Tables

**Figure 1 biomedicines-12-00596-f001:**
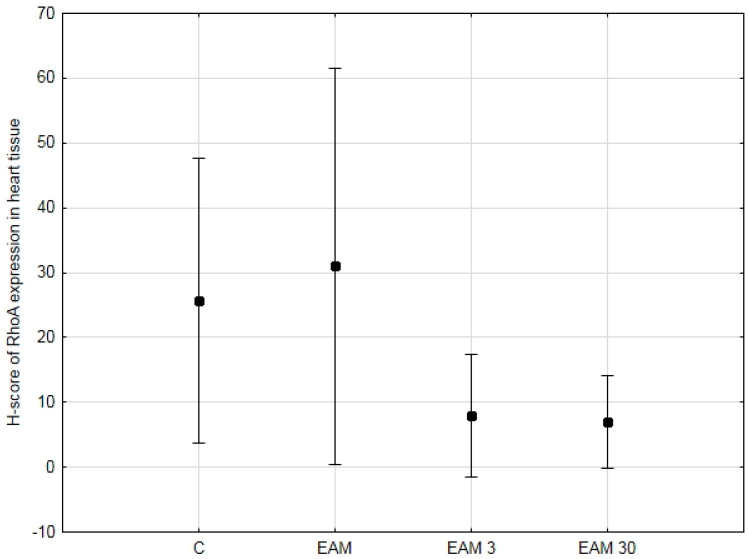
The comparison of the H-score of RhoA expression in heart tissue (means with 95% confidence intervals) between the groups. Evaluation of average H-score of RhoA expression in heart tissue revealed no significant differences between groups. C—control group (*n* = 8); EAM—rats with EAM (*n* = 5); EAM 3—EAM rats treated with simvastatin 3 mg/kg (*n* = 5); EAM 30—EAM rats treated with simvastatin 30 mg/kg (*n* = 5). ANOVA (with Welch’s correction): F(3, 9.66) = 2.32, *p* = 0.140, adj-R^2^ = 0.12. Variance homogeneity Levene’s test: F = 6.42, *p* = 0.003).

**Figure 2 biomedicines-12-00596-f002:**
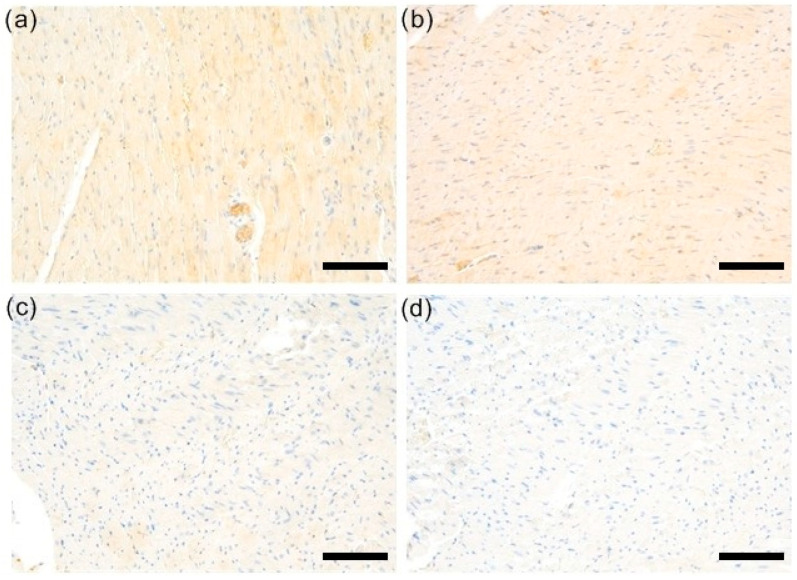
Representative samples of RhoA stains. (**a**) Weak RhoA expression was observed in C group. (**b**) Similarly, weak RhoA staining was typical for EAM cases. (**c**) EAM 3 group was characterized by barely perceptible or absent RhoA immunoexpression (**d**) EAM 30 group was characterized by barely perceptible or absent RhoA immunoexpression (magnifications: 200×, scale bars = 100 µm). C—control group (*n* = 8); EAM—rats with EAM (*n* = 5); EAM 3—EAM rats treated with simvastatin 3 mg/kg (*n* = 5); EAM 30—EAM rats treated with simvastatin 30 mg/kg (*n* = 5).

**Figure 3 biomedicines-12-00596-f003:**
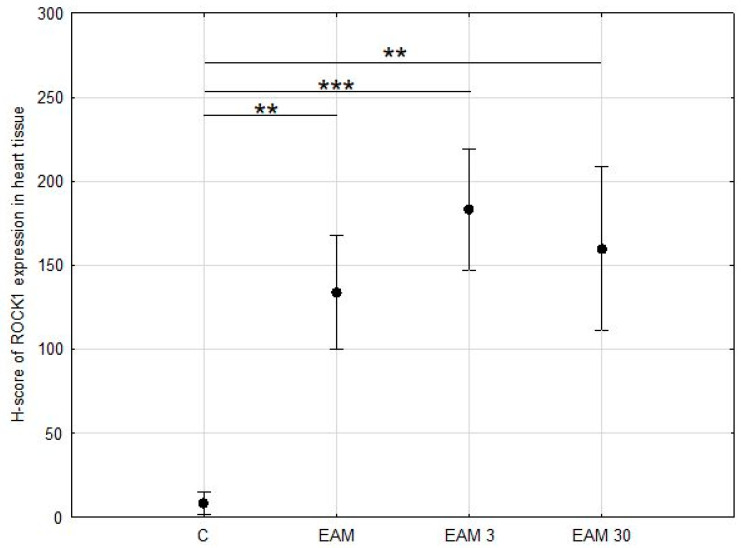
The comparison of the H-score of ROCK1 expression in heart tissue (means with 95% confidence intervals) between the groups. Evaluation of H-score of ROCK1 expression in heart tissue revealed a significant increase in ROCK1 expression in EAM groups. C—control group (*n* = 8); EAM—rats with EAM (*n* = 5); EAM 3—EAM rats treated with simvastatin 3 mg/kg (*n* = 5); EAM 30—EAM rats treated with simvastatin 30 mg/kg (*n* = 5). ANOVA: F(3, 19) = 60.83, *p* < 0.0001, adj-R^2^ = 0.89. Variance homogeneity Levene’s test: F = 2.05, *p* = 0.141. Post hoc Tukey’s test—C vs. EAM: *p* < 0.001; C vs. EAM 3: *p* < 0.001; C vs. EAM 30: *p* < 0.001; EAM vs. EAM 3: *p* = 0.036; EAM vs. EAM 30: *p* = 0.414; EAM 3 vs. EAM 30: *p* = 0.517. **—*p* < 0.01, ***—*p* < 0.001.

**Figure 4 biomedicines-12-00596-f004:**
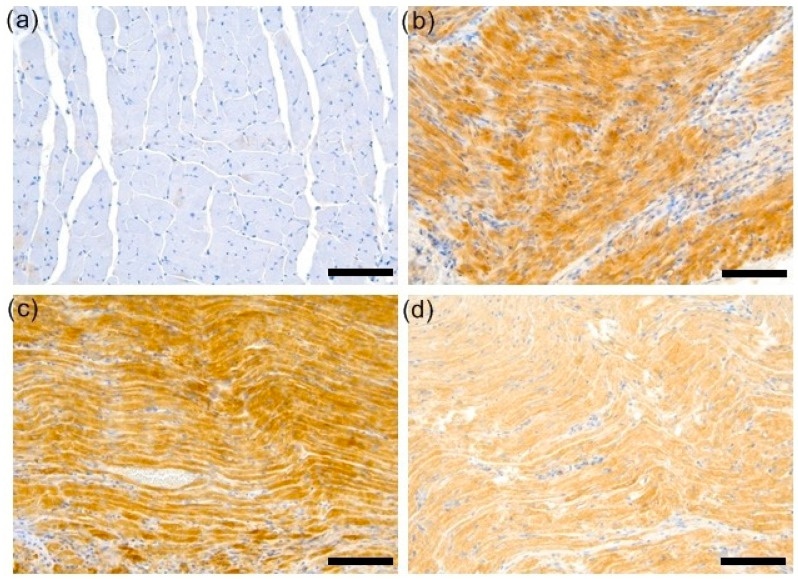
Representative samples of ROCK1 stains. (**a**) No appreciable ROCK1 staining in a section of a control specimen. (**b**) EAM cases were typified by strong ROCK1 expression in extensive tissue areas. (**c**) Representative sample from EAM 3 group showing diffuse ROCK1 staining of medium/strong intensity. (**d**) Representative sample from EAM 30 group showing diffuse ROCK1 staining of medium/strong intensity (magnifications: 200×, scale bars = 100 µm). C—control group (*n* = 8); EAM—rats with EAM (*n* = 5); EAM 3—EAM rats treated with simvastatin 3 mg/kg (*n* = 5); EAM 30—EAM rats treated with simvastatin 30 mg/kg (*n* = 5).

**Figure 5 biomedicines-12-00596-f005:**
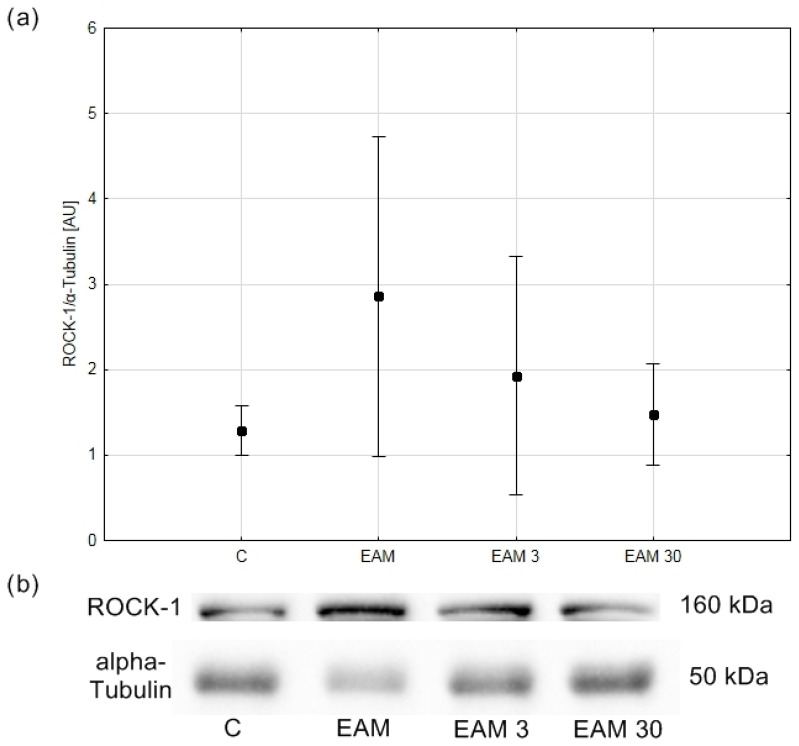
The comparison of ROCK-1/α-Tubulin (means with 95% confidence intervals) between the groups. (**a**) Western blot analysis revealed no significant differences in ROCK-1/α-Tubulin content between groups. C—control group (*n* = 4); EAM—rats with EAM (*n* = 5); EAM 3—EAM rats treated with simvastatin 3 mg/kg (*n* = 5); EAM 30—EAM rats treated with simvastatin 30 mg/kg (*n* = 5). ANOVA (with Welch’s correction): F(3, 7.84) = 1.87, *p* = 0.214, adj-R^2^ = 0.18. Variance homogeneity Levene’s test: F (3, 15) = 6.10, *p* = 0.006). (**b**) Representative blot of ROCK-1/α-Tubulin content in heart tissue. C—control group (*n* = 4); EAM—rats with EAM (*n* = 5); EAM 3—EAM rats treated with simvastatin 3 mg/kg (*n* = 5); EAM 30—EAM rats treated with simvastatin 30 mg/kg (*n* = 5).

**Figure 6 biomedicines-12-00596-f006:**
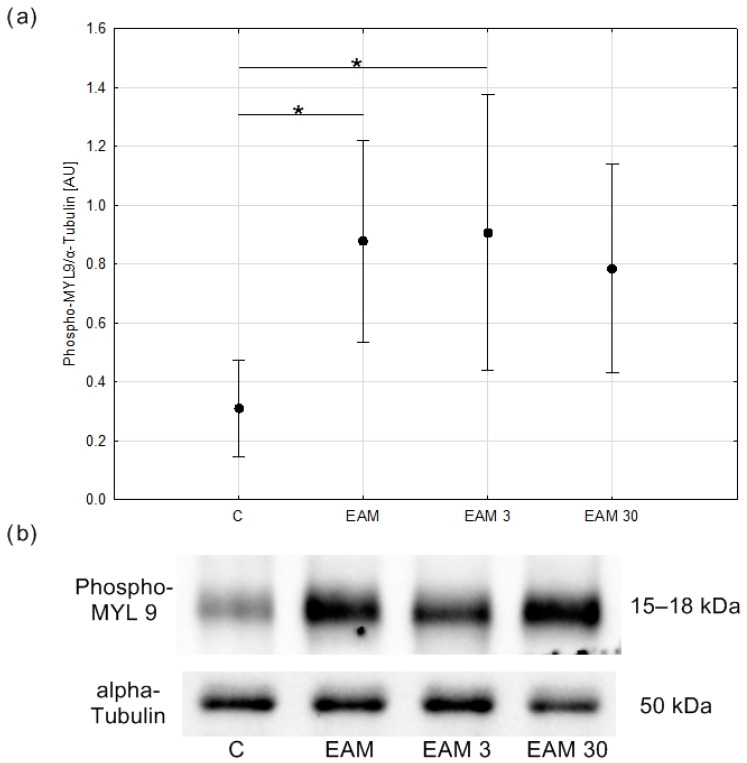
The comparison of Phospho-MYL9/α-Tubulin (means with 95% confidence intervals) between the groups. (**a**) Western blot analysis revealed an increase in phospho-MYL 9/α-Tubulin content in EAM groups being significant between C and EAM and C and EAM 3 groups. C—control group (*n* = 5); EAM—rats with EAM (*n* = 5); EAM 3—EAM rats treated with simvastatin 3 mg/kg (*n* = 5); EAM 30—EAM rats treated with simvastatin 30 mg/kg (*n* = 5). ANOVA: F(3, 16) = 4.87, *p* = 0.014, adj-R^2^ = 0.40. Variance homogeneity Levene’s test: F = 1.89, *p* = 0.172. Post hoc Tukey’s test—C vs. EAM: *p* = 0.027; C vs. EAM 3: *p* = 0.019; C vs. EAM 30: *p* = 0.073; EAM vs. EAM 3: *p* = 0.998; EAM vs. EAM 30: *p* = 0.954; EAM 3 vs. EAM 30: *p* = 0.901. *—*p* < 0.05. (**b**) Representative blot of phospho-MYL 9/α-Tubulin content in heart tissue. C—control group (*n* = 5); EAM—rats with EAM (*n* = 5); EAM 3—EAM rats treated with simvastatin 3 mg/kg (*n* = 5).

**Figure 7 biomedicines-12-00596-f007:**
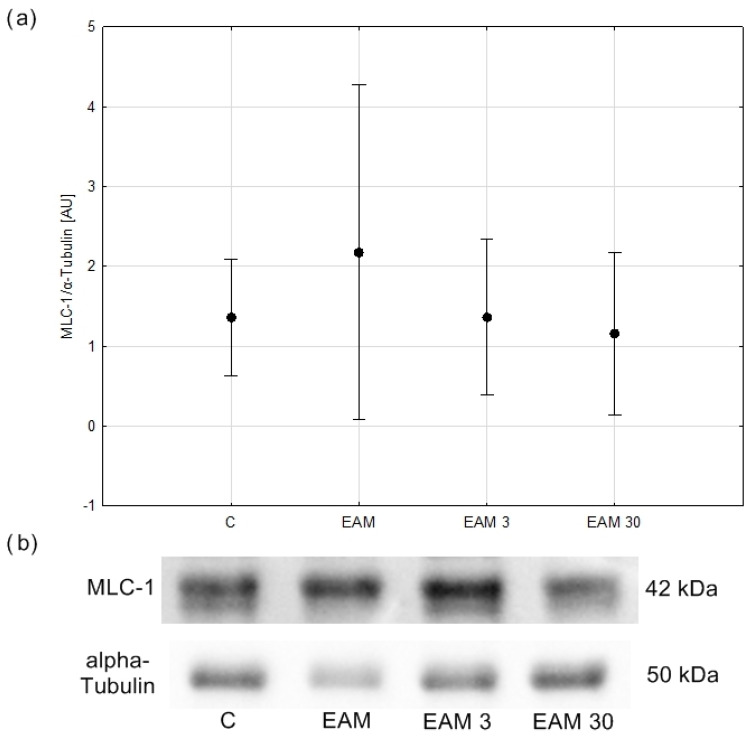
The comparison of MLC-1/α-Tubulin (means with 95% confidence intervals) between the groups. (**a**) Western blot analysis revealed no significant differences in MLC-1/α-Tubulin content between groups. C—control group (*n* = 5); EAM—rats with EAM (*n* = 5); EAM 3—EAM rats treated with simvastatin 3 mg/kg (*n* = 5); EAM 30—EAM rats treated with simvastatin 30 mg/kg (*n* = 5). ANOVA (with Welch’s correction): F(3, 8.58) = 0.430, *p* = 0.740, adj-R^2^ = −0.01. Variance homogeneity Levene’s test: F (3, 16) = 5.04, *p* = 0.012). (**b**) Representative blot of MLC-1/α-Tubulin content in heart tissue. C—control group (*n* = 5); EAM—rats with EAM (*n* = 5); EAM 3—EAM rats treated with simvastatin 3 mg/kg (*n* = 5).

## Data Availability

The data generated or analyzed during this study are included in this published article. Any datasets generated and analyzed during the current study and not included in this published article due to the presence of too many files or their size are available from the corresponding author upon request.

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
