# Peer review of "RhoA/ROCK Pathway Is Upregulated in Experimental Autoimmune Myocarditis and Is Inhibited by Simvastatin at the Stage of Myosin Light Chain Phosphorylation"

_biomedicines, 2024, doi:10.3390/biomedicines12030596_

Round 1

Reviewer 1 Report

Comments and Suggestions for Authors

This manuscript investigated the role of Simvastatin in Experimental Autoimmune Myocarditis (EAM), possibly through the regulation of MLC phosphorylation. However, the workload of the article is not substantial, the logic is somewhat confused, and I still have the following concerns:

1. My primary concern is that the important conclusion stated, "phosphorylated myosin light chain (phospho-MYL9) protein content was impacted by the highest dose of simvastatin," is not supported by the data. According to Fig 6, EAM rats treated with either 3 mg/kg or 30 mg/kg doses of simvastatin did not show a decrease in phospho-MYL9 levels compare with the EAM group.

2. Secondly, the entire manuscript does not provide a pathological description of EAM in rats, including data from rat cardiac ultrasound, heart weight, observation of inflammatory cell infiltration in cardiac sections, etc. This can not only demonstrate successful EAM modeling by the authors but also observe whether Simvastatin has a therapeutic effect on EAM in rats.

3. The third concern, as discussed by the authors in the limitations section, is that the manuscript does not delve into the association between MLC and MMP-9/MMP-2 in EAM disease model. Therefore, the logical inference in Lines 259-262 cannot be drawn. Supplementing the results in this section would make the manuscript more comprehensive.

4. All representative images of slices in the manuscript lack a scale bar.

5. The total number of rats mentioned in Line 109 is incorrect. According to the subsequent text, all EAM model rats should be 15 instead of 16.

6. According to the author's previously published article [10.3390/biom12091291] detecting RhoA protein levels should be feasible and can be supplemented in this manuscript.

7. As mentioned by the authors in the discussion section, the results of this manuscript differ from previous findings on RhoA/ROCK in EAM [doi:10.3390/cells9030700]. However, the authors only mentioned the differences without conducting a comparative analysis. Please add the author's understanding of the differences between the existing results and the results of this manuscript.

8. Line 353 seems incomplete at the end.

Author Response

We would like to thank the Reviewers for taking time and effort to review the manuscript. We are very grateful for all your valuable comments and suggestions that helped us further revise and improve the quality of the manuscript. Below, we detail the point-by-point responses to the Reviewers' comments. The relevant corrections are highlighted in the revised manuscript with track changes made of Word.

We hope that our revised manuscript is now suitable for recommendation to publication.

We are waiting for your feedback.

Reviewer 2 Report

Comments and Suggestions for Authors

Many studies have proven the involvement of the RhoA/ROCK pathway in autoimmune and cardiovascular diseases and the beneficial effects of its downregulation. Here, authors examined whether the effect of simvastatin on experimental autoimmune myocarditis (EAM) may be through targeting the Ras homolog family member A/Rho-associated coiled-coil containing kinases (RhoA/ROCK) pathway and whether previously shown downregulation of metalloproteinase 9 (MMP-9) could be associated with MLC phosphorylation. Two doses of simvastatin were administered to experimental rats with autoimmune myocarditis by gastric gavage for 3 weeks. Immunohistochemical staining for RhoA and ROCK1 was evaluated semi-quantitatively with H-score. The RhoA staining showed no significant differences in expression between the groups. However, the ROCK1 expression was significantly upregulated in the hearts of the EAM group and was not downregulated by simvastatin. The Western blotting analysis of the last downstream product of the RhoA/ROCK axis, phosphorylated myosin light chain (phospho-MYL9) revealed that protein content increased in EAM hearts and was downregulated by the highest dose of simvastatin. These findings suggest that the RhoA/ROCK pathway is upregulated in EAM, and simvastatin in EAM settings inhibits the RhoA/ROCK pathway at the stage of phosphorylation of myosin light chains and provide a new insight into the molecular pathology of autoimmune myocarditis.

These results, although preliminary, are of great importance as they represent one of the first attempts to assess the role of the RhoA/ROCK pathway in the pathogenesis of EAM.

The article is well-written, didactic, and a starting point for new experiments to explain the pathophysiology of autoimmune myocarditis. I recommend completely revising the English used to make the sentences more streamlined and understandable.

Comments on the Quality of English Language

I recommend completely revising the English used to make the sentences more streamlined and understandable.

Author Response

(The authors gave the same response as above.)

Reviewer 3 Report

Comments and Suggestions for Authors

The study investigates the role of the RhoA/ROCK pathway in experimental autoimmune myocarditis (EAM), assessing the effect of simvastatin on this pathway. It was found that while RhoA expression remained unchanged, ROCK1 expression was significantly upregulated in EAM and not affected by simvastatin. However, simvastatin was able to downregulate the phosphorylation of myosin light chains (MLC), a downstream product of the RhoA/ROCK pathway, suggesting a potential therapeutic mechanism in EAM.

1. Provide more details on the dose rationale for simvastatin, including references to previous studies or pharmacokinetic data supporting the chosen doses.

2. Expand on the methods used for immunohistochemical staining and Western blot analysis, including controls, to ensure reproducibility.

3. Address the variability and reliability of the Western blot data, particularly for ROCK1, where the results were not statistically significant.

4. Consider including additional molecular markers to support the findings, offering a broader understanding of the pathway's regulation.

5. Explore the potential effects of simvastatin on other pathways involved in EAM, providing a more comprehensive view of its therapeutic action.

6. Discuss the study's limitations, including the translation of findings from an animal model to human disease.

7. Provide a more detailed discussion on the implications of these findings for treating autoimmune myocarditis, considering current treatment limitations and the potential role of simvastatin.

Author Response

(The authors gave the same response as above.)

Round 2

Reviewer 1 Report

Comments and Suggestions for Authors

Thank authors for the response and revisions. I am satisfied with the responses and changes, which have addressed my concerns adequately. Hoping the enriched and reliable discussion section will provide valuable insights for other researchers.

Reviewer 3 Report

Comments and Suggestions for Authors

After revision, the manuscript improved and the author responded to the comments clearly. Thanks for the opportunity to review this work.